



# U-Plume: Automated algorithm for plume detection and source quantification by satellite point-source imagers

Jack H. Bruno[1,2], Dylan Jervis[2], Daniel J. Varon[1], Daniel J. Jacob[1]

[1]Earth And Planetary Sciences, Harvard University, Cambridge, MA, 02138, USA
[2]GHGSat Inc, Montreal, Quebec, H2W1Y5, Canada

*Correspondence to*: Jack H. Bruno (jackbruno@g.harvard.edu)

**Abstract.** Current methods for detecting atmospheric plumes and inferring point source rates from high-resolution satellite imagery are labor intensive and not scalable to the growing satellite dataset available for methane point sources. Here we

present a two-step algorithm called U-Plume for automated detection and quantification of point sources from satellite imagery. The first step delivers plume detection and delineation (masking) with a machine learning U-Net architecture for image segmentation. The second step quantifies point source rate from the masked plume using wind speed information and either a convolution neural network (CNN) or a physics-based Integrated Mass Enhancement (IME) method. The algorithm can process 62 128×128 images per second on a single core. We train the algorithm with large-eddy simulations of methane plumes

superimposed on noisy and variable methane background scenes from the GHGSat-C1 satellite instrument. We introduce the concept of point source observability $O_{ps} = Q/(UW\Delta B)$ as a single dimensionless number to predict plume detectability and source rate quantification error from an instrument as a function of source rate $Q$, wind speed $U$, instrument pixel size $W$, and instrument-dependent background noise $\Delta B$. We show that $O_{ps}$ can powerfully diagnose the ability of an imaging instrument to observe point sources of a certain magnitude under given conditions. U-Plume successfully detects and masks plumes from

sources as small as 100 kg h[-1] over surfaces with low background noise and succeeds for larger point sources over surfaces with substantial background noise. We find that the IME method for source quantification is unbiased over the full range of source rates while the CNN method is biased toward the mean of its training range. The total error in source rate quantification is dominated by wind speed at low wind speeds and by the masking algorithm at high wind speeds. A wind speed of 2-4 m s[-1] is optimal for detection and quantification of point sources from satellite data.


Keywords: Methane, Point Sources, Satellites, Machine Learning, Image Segmentation

## 1 Introduction

A number of satellite instruments can now detect and image methane column plumes with spatial resolution finer than 50 m by solar backscatter in the shortwave infrared (SWIR), enabling quantification of large point sources from individual facilities

(Jacob et al., 2022). As the satellite observing system expands for both methane and other gases, there is a growing need for efficient methods of detecting and quantifying these point sources by automated processing of vast amounts of data. Here we



present the U-Plume algorithm, a generalized machine-learning method to address this need, and we apply it to methane observations from GHGSat (Jervis et al., 2021).

Several methods have been proposed for inferring point source rates from satellite imagery of atmospheric pollution plumes, including for methane (Varon et al. 2018), $CO_2$ (Nassar et al. 2017), $NO_2$ (Valin et al. 2013; De Foy et al. 2015; Beirle et al. 2021), CO (Pommier et al. 2013), $SO_2$ (Fioletov et al. 2015; McLinden et al. 2016), and $NH_3$ (Clarisse et al. 2019; Dammers et al. 2019; Noppen et al. 2023). Gaussian plume inversion (Bovensmann et al., 2010; Krings et al., 2011, 2013) and mass balance methods (Jacob et al. 2016; Buchwitz et al., 2017) have both shown success in estimating emissions for very large

plumes (>10 km) but they fail when applied to sub-km plumes due to stochastic turbulence that leads to non-Gaussian behavior and dominance of eddy flow (Varon et al., 2018). Temporal averaging of plumes over multiple satellite passes with wind rotation has been used extensively for large plumes from well-identified point sources to decrease noise and enhance Gaussian behavior (Pommier et al. 2013; Fioletov 2015; McLinden et al. 2016; Varon et al., 2020; Massakers et al. 2022), but is difficult to implement for methane plumes that are typically smaller and intermittent (Frankenberg et al., 2016; Cusworth et al., 2021).


Two methods that have shown success in estimating emissions from high-resolution instantaneous plume imagery are the Integrated Mass Enhancement (IME) and Cross-Sectional Flux (CSF) methods (Krings et al. 2011, 2013; Frankenberg et al. 2016; Varon et al. 2018). The IME method relates the total observed plume mass to a source rate, using information from the plume size and the local wind speed. The CSF method integrates concentrations over plume cross-sections perpendicular to

the wind direction and multiplies them by the local wind speed to infer a source rate. Both methods require the plume to be identified and masked (i.e., delineated) within the image.

Detection and masking of plumes in satellite scenes has generally been done by human analysts (Guanter et al., 2021) but this is not practical operationally. Simple statistical thresholding approaches combined with adjacency criteria have been developed

to detect methane enhancements above background for plume masking (Varon et al., 2019; Duren et al. 2019) but these are vulnerable to retrieval artifacts, particularly from surface features mistakenly identified as methane plumes (Cusworth et al., 2019).

Machine learning is a promising avenue for automating plume detection and inference of point source emissions.

Jongaramrungruang et al. (2022) proposed a Convolutional Neural Network (CNN) to estimate source rates directly from aerial AVIRIS-NG methane imagery. Their method (called MethaNet) provides source rate estimates based solely on methane column retrieval fields, forgoing the use of external information on wind speed by exploiting wind information contained in the morphology of the plumes (Jongaramrungruang et al., 2019). Training and testing of MethaNet has been done on aircraft imagery with relatively clean backgrounds and high pixel resolution (3 m), but to our knowledge it has not been applied to

satellite imagery which is coarser and more vulnerable to surface artifacts. MethaNet does not produce a plume mask (outline



of plume boundaries), but such a mask offers important visual information and can help identify artifacts. Joyce et al. (2023) used a U-Net architecture to mask methane plumes in PRISMA satellite imagery directly from radiances and again without wind speed information. Separate networks were then used for inferring methane column concentration and estimating emission rates. In both MethaNet and the Joyce et al. (2023) approach there is a bias towards the training mean in the final

source rate estimates, resulting in overestimate of low emitting plumes and underestimate of high emitting plumes. This is a common issue in machine learning applications.

Here we perform binary segmentation of methane column imagery from GHGSat using a U-Net neural network architecture to produce plume masks, and we then use these masks together with wind speed information to infer point source rates either

with a CNN or with the physics-based IME method. We use wind speed information because some is always available, either from local measurements or from regional/global reanalysis datasets. The U-Net architecture (Ronneberger et al. 2015) has previously demonstrated to be effective in feature recognition by binary image segmentation from satellite imagery (Joyce et al. 2023), avoiding the false positives present in more traditional methods reliant on thresholding (Rezvanbehbahani et al. 2020). Combining the U-Net architecture to detect plumes in column enhancement with subsequent steps to infer point source

rates forms an end-to-end algorithm that we call U-Plume.

## 2 Methods

We present here the U-Plume algorithm for identifying point source plumes in satellite imagery of methane concentrations and inferring point source rates. The algorithm has two components: (1) a U-Net to detect plumes by binary segmentation of the pixels in the satellite scene (0 = background, 1 = plume), producing a plume mask; (2) two alternative CNN and IME

approaches for inferring source rates from the plume mask. We demonstrate U-Plume on the GHGSat-C1 instrument for methane, but the architecture and training process presented here are applicable to any satellite point source imagers for any species.

GHGSat-C1 is the first operational instrument launched in 2020 by GHGSat Inc., following the demonstration instrument

GHGSat-D launched in 2016 (Jervis et al., 2021). The instrument is a shortwave interferometer with an observation domain of ~12×12 km$^2$ at 25×25 m$^2$ pixel resolution. Retrieval of the backscattered solar spectrum at the 1.65 μm methane absorption band yields an estimated methane column mass enhancement over background reported as mol m$^{-2}$ with 1-2% estimated precision. The instrument is in polar sun-synchronous orbit with observation at ~9:30 am local time. Subsequently launched GHGSat-C2-C8 instruments have less background noise than -C1. Our results can be extended to C2-C8 and any other

instruments using the background noise metrics presented below.

Our first step is to create a training data set of plume-containing satellite images in which synthetic instantaneous plumes of known source rates are superimposed on actual plume-free images. Following Varon et al. (2018) and Jongaramrungruang et





al. (2019), we use the Weather Research and Forecasting Model Large Eddy Simulation (WRF-LES) to create a diverse plume
dataset of atmospheric methane concentration enhancements where the relationship between source rate and plume
concentrations is known. Point source methane emission and atmospheric transport are simulated on a 25×25 m$^2$ horizontal
and 15-m vertical grid over a 9×9×2.4 km$^3$ domain, with the point-source emitting at the surface 5/6 from the downwind edge
of the domain. We conduct 4 simulations for different meteorological conditions, each run for 3 hours with the first hour used
as spin-up, resulting in 2 hours of usable plume images per simulation. Time steps for WRF-LES integration are 1/4 second
and instantaneous plumes are sampled every 30 seconds. Mean wind speeds for the simulations are 3-9 m s$^{-1}$ with sensible heat
fluxes of 100-300 W m$^{-2}$ and mixed layer depths of 500-2000 m. This is the same ensemble as used by Varon et al (2021).

To create an effective neural network, the training imagery must be as close as possible to the real observations that the model
will be applied to. Previous work by Varon et al. (2018) using WRF-LES to simulate synthetic plumes superimposed these
plumes on a white-noise background to generate a training data set, but variable surface albedo and terrain can lead to
heterogenous noise fields with complex structure. To create realistic noise for our training images we start from a set of actual
GHGSat-C1 observations of plume-free scenes and add the methane column enhancements from the WRF-LES simulation.
We use the standard deviation of the pixel enhancements in the plume-free scene ($\Delta B$, kg m$^{-2}$) as measure of background
noise:


$$\Delta B = \sqrt{\frac{\sum_{i=1}^{n}(x_i - \bar{x})^2}{n}} \tag{1}$$

Here $n$ is the number of pixels in the image, $x_i$ is the column concentration for individual pixel $i$, and $\bar{x}$ is the mean column
concentration in the scene. When discussing $\Delta B$ we express it as a percentage of the global mean background concentrations
(taken as $x_b = 0.011$ kg m$^{-2}$; Jervis et al., 2021). We use 28 12×12 km$^2$ GHGSat-C1 plume-free observations corresponding to
a variety of surface types. Scenes with $\Delta B < 5\%$ correspond to homogenous bright surfaces including arid and grassland
terrain. Scenes with $5\% < \Delta B < 10\%$ correspond to moderately heterogeneous surfaces. Scenes with $\Delta B > 10\%$ correspond
to very heterogeneous or dark surfaces including forests, wetlands, and urban areas. Values of $\Delta B$ are instrument dependent.
The more recent GHGSat-C2 and beyond instruments have $\Delta B < 2\%$ and across all scenes and observing conditions (Ramier,
2022). On the other hand, hyper/multi-spectral land surface mappers can have background noise exceeding 10% (Cusworth et
al., 2019; Varon et al., 2021).



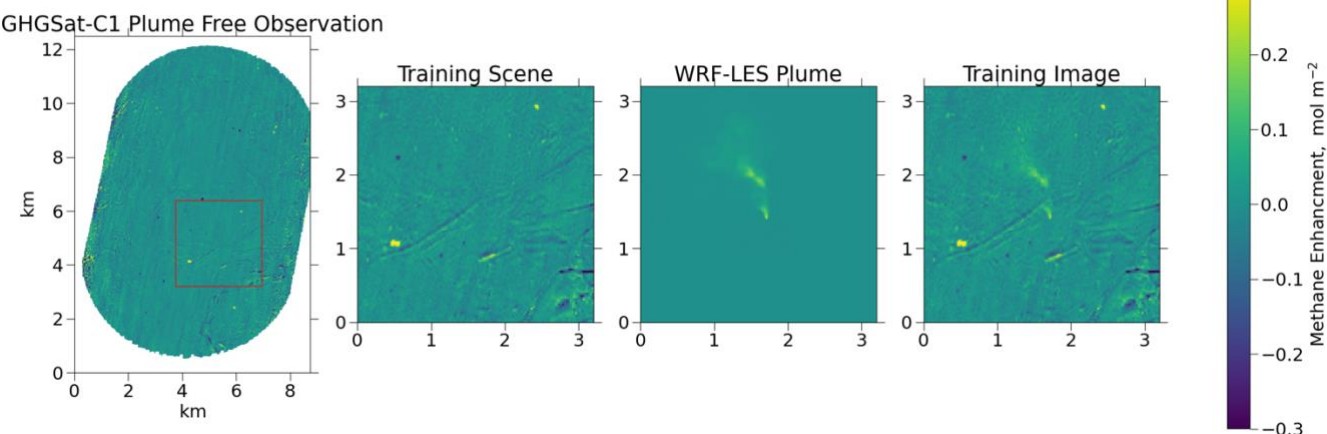

**Figure 1: Sample generation of U-Plume training imagery. The first panel shows a sample plume-free observation of methane column concentration enhancements observed by GHGSat-C1. Panel 2 is a 128×128-pixel scene randomly selected from the full observation domain. Panel 3 shows a sample WRF-LES instantaneous plume for a source rate of 2000 kg h⁻¹ (we choose a large plume here for easy visualization). Panel 4 shows the result of adding panels 2 and 3 to form a training image. 0.1 mol m⁻² corresponds to a dry column mixing ratio enhancement of 284 ppb.**

Fig 1 illustrates the steps for creating the synthetic plume observations used in training and testing our method. For each training image, a 128×128-pixel scene is randomly selected from a plume-free GHGSat-C1 observation. A random plume from the WRF-LES dataset is then selected, with a random rotation and translation applied to situate the plume anywhere in the scene but with a safeguard to ensure that the plume is fully contained in the scene. Since the methane enhancement in the plume is proportional to source rate, we can scale the plume randomly to correspond to a given source rate. This plume image is then added to the selected scene to create a training image. A training range of plumes emitting in the 500-2000 kg h⁻¹ is chosen to ensure that the plumes are at least partially visible in most training images. Though we want the network to be able to detect smaller source rates, it is important that the structural features being searched for are visible above noise within the training dataset. Because the U-Net recognizes structure, not just enhancement, the network trained on the 500-2000 kg h⁻¹ range is still usable outside of this range as we will see.

Our set of training/test images is created using 28 plume-free observations. We step the 128×128-pixel scene in 16-pixel steps to create a set of scenes. The $\Delta B$ values of these scenes range from 1% to more than 50%. We only use scenes with $\Delta B < 20\%$ to avoid training on scenes with extremely high background variability where detection of any but the largest plumes would be intractable. This filtering leaves us with a total of 6870 scenes. We then apply the random plume placement and source magnitude as described above to create 6870 simulated plume observations. The true plume mask is defined as pixels where the enhancement from the LES simulation is higher than the $\Delta B$ of the scene, which is the criterion for the pixel contributing more information than noise (Varon et al., 2018). 90% of the images are used for training and 10% are set aside for testing.



Fig. 2 shows the two-step process used to obtain source rate estimates from methane enhancement images. First, a U-Net
masking network is used to identify pixels in which a plume is present. The binary mask created by this network is then fed
along with the original enhancement image and external wind speed data to estimate the source rate from the identified plume.
The enhancement image with mask is preserved for visualization and as a quality control diagnostic. Processing the test dataset
of 687 images on a single 2.6 GHz Intel Core i7 CPU takes 11 seconds, corresponding to 62 images per second.

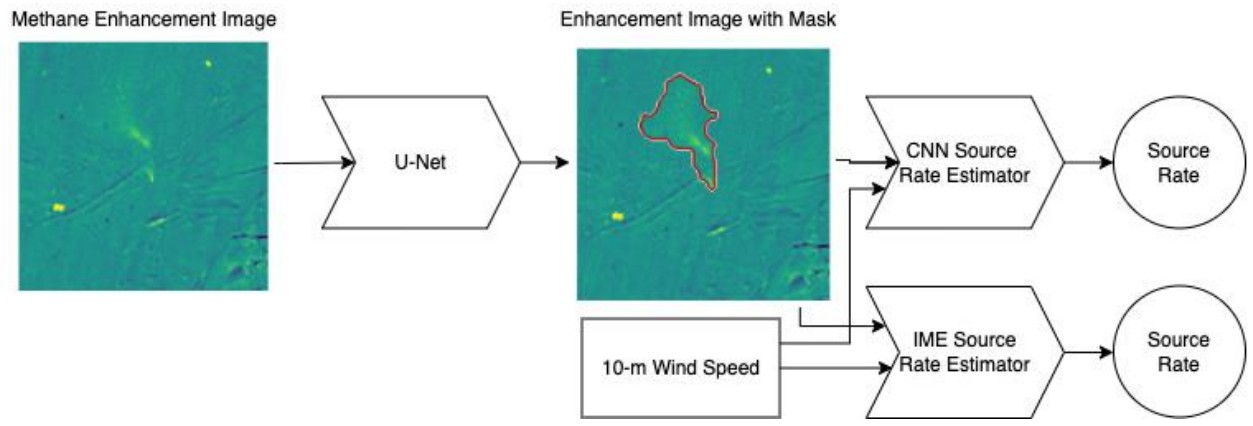


**Figure 2: U-Plume architecture. Starting from a methane enhancement image (the sample image of fig.1), we use the U-Net machine
learning algorithm to produce a plume mask (in red) of pixels containing more information than noise. All masked pixels in a single
128×128-pixel (3.2×3.2 km²) scene are assumed to originate from the same plume. We then use the original enhancement image, the**
**mask, and local 10-m wind speed to quantify emissions using either a machine learning method or an Integrated Methane
Enhancement (IME) method.**

The U-Net neural network for plume masking (Ronneberger et al. 2015) uses an encoder-decoder framework to recognize
structural features within imagery. The network receives a methane enhancement image as in Fig. 2 and classifies each pixel
with a confidence level for the presence of a plume. We apply a threshold of 50% confidence to produce a binary plume mask.
The main advantage of this method over traditional threshold masking is that high enhancement features that do not have
plumelike shape characteristics will not be falsely captured as plumes.

The encoder portion of the network applies a series of convolutional filters and pooling layers to identify features of increasing
complexity within the image. At the end of the encoding the image has been compressed spatially and has many channels
corresponding to the various feature maps created by the convolutional filtering. By compressing the image spatially in this
way, the network can recognize connections between distant features.



The decoder portion of the network serves to interpret these identified features as criteria for classification. An inverse
convolutional operator is used to expand the spatially compressed image back to the original size of the input image while
reducing the number of filters applied, which compresses the number of channels in the image. The final step of the network
uses the information gathered from the encoder-decoder process to estimate a confidence level (0-1) that each pixel is or is not
part of a plume. Output from this model for a single image is a confidence mask for plume location across the enhancement
image as shown in Figure 2. In practice. the confidence mask is generally either very close to 1 or very close to 0. Intermediate
values fall off very quickly at the edge of the mask.

Once the U-Net mask has been generated as the first step of our U-Plume algorithm, we infer the source rate as a second step
using as options either a CNN method or an IME method. The CNN method takes in a 3-channel input image containing the
original image, the binary mask, and 10-m wind speed. Initial testing indicated that including wind direction did not improve
performance. The wind speed channel is populated with a uniform 15-minute scene average value from the LES simulation.
The CNN uses a series of convolutional and pooling layers which connect to a set of fully connected layers, rather than a
decoder, to yield a scalar estimate of emissions.

The IME method follows Varon et al. (2018), in which the plume mass enhancement (IME) is combined with a parameterized
effective wind speed ($U_{eff}$) and plume length scale ($L$) to infer the source rate ($Q$, kg h$^{-1}$):

$$Q = \text{IME} \frac{U_{eff}}{L} \tag{2}$$

The plume length scale $L$ is defined as the square root of the plume mask area. The effective wind speed is fitted to the 15-
minute averaged wind speed $U_{10}$ at 10-m altitude using the training data set where the relationship between $Q$, IME, and $L$ is
known for a given $U_{10}$. This yields

$$U_{eff} = 0.7 + 0.23\, U_{10} \tag{3}$$

with $U_{eff}$ and $U_{10}$ in units of m s$^{-1}$. The intercept of 0.7 m s$^{-1}$ arising from the fit can be physically interpreted as a minimum
turbulent diffusion for the plume at low wind speeds.

**3 Dimensionless Point Source Observability Number**

We introduce the concept of point source observability ($O_{ps}$) as a dimensionless number to determine the ability of a satellite
instrument to detect and quantify point sources. Observability is a function of source rate ($Q$, kg s$^{-1}$), wind speed ($U$, m s$^{-1}$),
instrument pixel resolution ($W$, m), and scene-dependent background noise ($\Delta B$, kg m$^{-2}$, as defined by equation (1)).



The idea behind the $O_{ps}$ concept is that the observability of point sources is determined by the signal-to-noise ratio in the plume (Varon et al., 2018). The column concentration enhancement in the plume (kg m$^{-2}$) scales as $Q/UW$ (Jacob et al., 2016). The noise for a given scene (kg m$^{-2}$) is given by $\Delta B$ which is a scene-dependent property of the instrument that can be characterized as a function of surface type. $\Delta B$ can be viewed as the precision of the instrument accounting for the contribution from surface artifacts. The dimensionless point source observability number $O_{ps}$ is then given by

$$O_{ps} = \frac{Q}{UW\Delta B} \tag{4}$$

and is a measure of the signal-to-noise ratio. We will use it here to interpret our results, but it can be applied more generally to any remote sensing observation of point sources.

## 4 Results

### 4.1 Plume Detection and Masking


Fig. 3 shows the relationship between source rate and U-Net masking success. The metric we use for masking success is the Jaccard score, which is the intersection of the predicted mask (A) and true mask (B) over the unity of the two:

$$J(A,B) = \frac{A \cap B}{A \cup B} \tag{5}$$


In 2% of cases no mask is produced for the image, and in 30% of cases there is no overlap between the true and predicted masks. These failures occur for high background noise ($\Delta B < 5\%$) and lower emission rates. On the other hand, Jaccard scores generally exceed 0.5 for low background noise ($\Delta B < 5\%$).




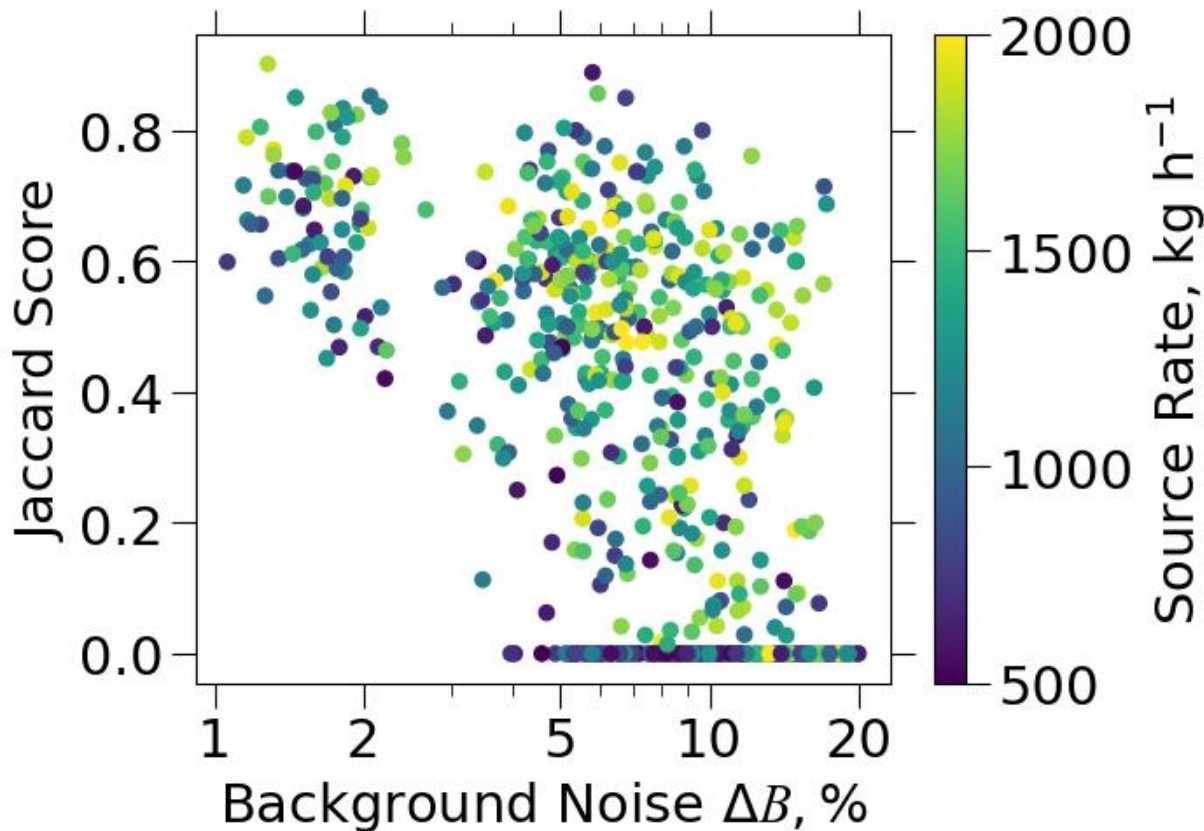

**Figure 3: Ability of the U-Net machine learning algorithm to delineate (mask) plumes emitted from point sources. Masking success is measured by the Jaccard score (equation (5)) and is plotted as a function of background noise $\Delta B$ (equation (1)) for a range of point source rates from 500 to 2000 kg h⁻¹. Results are for the test set of WRF-LES plume images superimposed on noisy plume free observations from the GHG-Sat C1 satellite instrument.**

We examine false positives by applying the network over the test dataset of backgrounds without plumes added. This yields

masks in 14% of cases (false positive detection rate). The masks produced in these false positives are generally very small, with a median size of 5 pixels. Applying a minimum 5-pixel mask size brings the false positive rate down to 6% while losing only 1.6% of the true positive detections in the test data set and removing 92% of the zero overlap masks.




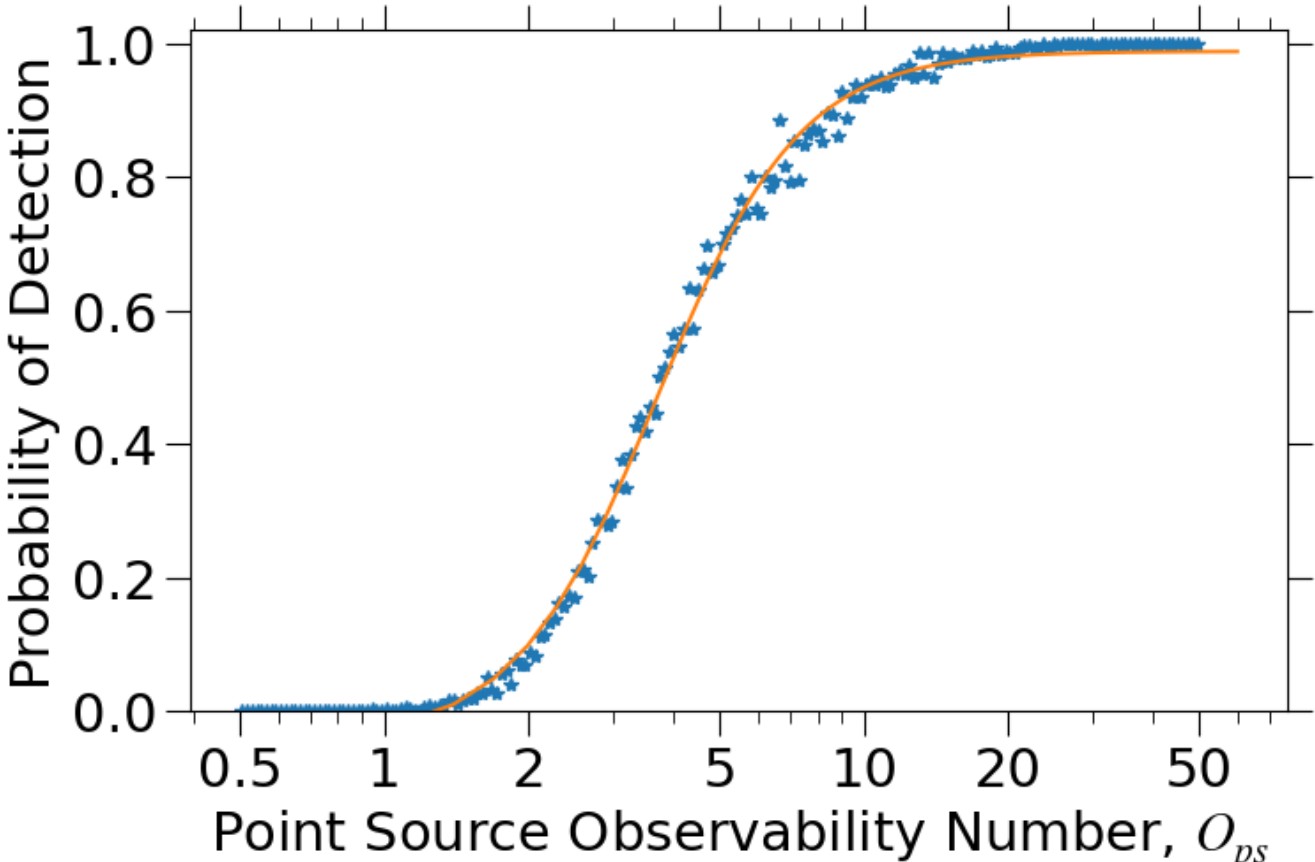


**Figure 4: Probability of detection ($J > 0.1$) for point sources in the GHGSat-C1 augmented test data set as a function of the point source observability dimensionless number $O_{ps}$ (equation (4)). Calculation of $O_{ps}$ uses the WRF-LES 10-m wind speed for $U$ and the GHGSat-C1 pixel size $W = 25$ m.**

The point source observability ($O_{ps}$) in equation (4) provides a scaling dimensionless number to better understand how point source detection for a given instrument relates to the combination of source rate, background noise, wind speed, and pixel size. Here we use the 10-m wind speed ($U_{10}$) in the calculation of $O_{ps}$ We create an augmented test dataset with 10 subset images from each of the 28 plume-free observations (Figure 1) and 20 emission levels for each image from 100 to 2000 kg h$^{-1}$ in 100 kg h$^{-1}$ increments. We bin images in the augmented test dataset by their $O_{ps}$ ranging from 0.5 to 50 in 201 log-spaced bins, and

for each bin with a minimum of 50 images we calculate the probability of detection as the fraction of plumes in the bin with $J > 0.1$. Figure 4 shows the relationship between the probability $P$ of detection and $O_{ps}$. The relationship for $O_{ps} > 1.4$ can be tightly fit to a sigmoid function of the form:

$$P = \frac{1.03}{1 + e^{-2.9(log(O_{ps}) - 1.3)}} - 0.05 \qquad (O_{ps} > 1.4) \qquad (6)$$




Detection probability is 10% for $O_{ps} = 0.2$, 50% for $O_{ps} = 4$, and 90% for $O_{ps} = 8$. The transition from low detectability ($<$10%) to high detectability ($>$90%) is very sharp in $O_{ps}$ space, demonstrating the power of the $O_{ps}$ metric for evaluating the ability of plume imaging instrument to observe point sources. The value $O_{ps} = 4$ for the 50% detection threshold can be interrpreted as a characteristic concentration enhancement $Q/4UW$ in the plume. The maximum concentration enhancement in the source pixel would be $Q/UW$ (Jacob et al., 2016).





**Figure 5: Probability of detection ($J > 0.1$) for point sources calculated using equation (6) as a function of background noise (equation (1)), source rate (top panel), and 10-m wind speed (bottom panel).**




Fig. 5 shows the probability of detection calculated using equation (6) as a function of source rate, 10-m wind speed, and background noise. For low values of $\Delta B$, the probability of detection is very sensitive to changes in $Q$ and less sensitive to

changes in $U_{10}$. With $\Delta B = 1\%$, 90% probability of detection is achieved for $Q > 400$ kg h$^{-1}$. Conversely, for high values of $\Delta B$ the probability of detection is very sensitive to changes in $U_{10}$ and less sensitive to changes in $Q$.

### 4.2 Source Rate Estimation



**Figure 6: Source rate estimates from the alternative CNN and IME methods (Figure 2) for point sources emitting in the 100-2000 kg h$^{-1}$ range (discrete 100 kg h$^{-1}$ bins) in the augmented test dataset. The methods were trained with source rates in the 500-2000 kg h$^{-1}$ range. Only images with Jaccard scores > 0.1 are processed. Error bars show 1 standard deviation of estimates.**

Fig. 6 shows the source rate estimates for the complete U-Plume workflow, starting from the plume imagery and applying either the CNN or the IME method for source rate quantification. We restrict our analysis to plumes with Jaccard score >0.1 to enable generalization to instruments with less background noise than GHGSat-C1 (such as GHGSat-C2) or to the use of improved filtering such as requiring a minimum number of plume pixels. Results are for the augmented dataset covering the 100-2000 kg h$^{-1}$ range but trained only over the 500-2000 kg h$^{-1}$ range. We assume for now no error on wind speed.


The CNN method has a smoothing bias where low source rates are overestimated, and high source rates are underestimated. This smoothing towards the mean is a common issue in machine learning methods and appears in the work of Jongaramrungruang et al. (2022) and Joyce et al. (2023). In addition, the CNN method is unable to extrapolate outside its training range. By contrast, the physically based IME method can successfully fit source rates all the way down to 100 kg h$^{-1}$

as long as the plumes are detectable. Discussion from this point forward will focus on the IME method as performing better than the CNN method over the full range of source rates. We retain the CNN method as an option in the U-Plume algorithm because it offers an alternative source rate estimate for verification purposes and because it could be improved in the future with a more extensive training dataset.

**4.3 Error Analysis**

The uncertainty on source rate estimates using U-Plume with the IME method can be expressed as the relative error standard deviation ($\sigma_Q$) of the U-Plume estimate versus the true value. It has two independent components: (1) error in the U-Plume algorithm, including the masking process and the IME parameterization ($\sigma_{Q,M}$); and (2) error in the 10-m wind speed ($\sigma_{Q,U}$). Here we derive expressions for $\sigma_{Q,M}$ and $\sigma_{Q,U}$ and add them in quadrature to quantify the overall error and determine the

driving factors.

The relative error standard deviation $\sigma_{Q,M}$ from the U-Plume algorithm can be parameterized using the augmented test data set binned by point source observability $O_{ps}$ following the approach in section 4.2 using only bins with >50 positive detections ($J$ > 0.1). Figure 7 shows the relative error standard deviation plotted against $O_{ps.}$






**Figure 7: Relative error on source rate estimation by the U-Plume algorithm. The figure shows the relative error standard deviation $\sigma_{Q,M}$ versus the point source observability dimensionless number $O_{ps}$ (equation (4)). Errors are calculated for point sources emitting in the 100-2000 kg h$^{-1}$ range (discrete 100 kg h$^{-1}$ bins) in the augmented test dataset, not including error in wind speed (for which the error is calculated separately, see text). Errors from individual images are binned by values of their point source**
**observability (equation (4)) for $O_{ps}$ = 0.5-50 in 201 log-spaced intervals. The orange line is a reduced-major-axis (RMA) regression fit (R$^2$ = 0.89) for $\sigma_{Q,M}$ versus $O_{ps}$ (equation (7)).**

Fitting the error to log ($O_{ps}$) yields:





$$\sigma_{Q,M} = \max(0.1, (0.52 - 0.12 \log(O_{ps})) \quad (O_{ps} > 2) \tag{7}$$

For values of $O_{p\,s} < 2$ the probability of detection is less than 10% (section 4.1). The relative error standard deviation bottoms out at 10% for highly observable plumes ($O_{ps} > 30$), reflecting irreducible error inherent to the IME method.

Uncertainty in wind speed adds another source of error. Let $\sigma_U$ (m s$^{-1}$) denote the error standard deviation in the 10-m wind speed $U_{10}$. Replacing in equations (2) and (3) yields a corresponding relative error standard deviation $\sigma_{Q,U}$ on the inferred IME source rate $Q$:

$$\sigma_{Q,U} = \frac{0.23\ \sigma_U}{0.7 + 0.23\ U_{10}} \tag{8}$$

Wind speed data may be available locally or from meteorological analysis datasets. A global default is the NASA Goddard Earth Observation System – Fast Processing (GEOS-FP) data publicly available on a 0.25° × 0.3125° grid at 1-h temporal resolution. Varon et al. (2018) estimated an error standard deviation $\sigma_U = 2$ m s$^{-1}$ for the GEOS-FP data, independent of wind speed magnitude, by comparison to 5-minute observations at US airports. Replacing into equation (8) indicates a maximum relative error standard deviation of 66% at low wind speed, decreasing with increasing wind speed. At high wind speeds the error becomes small but the IME is then small so that the masking error is large as inferred from the inverse dependence of the point source observability on wind speed (equation (4) and Figure 7).

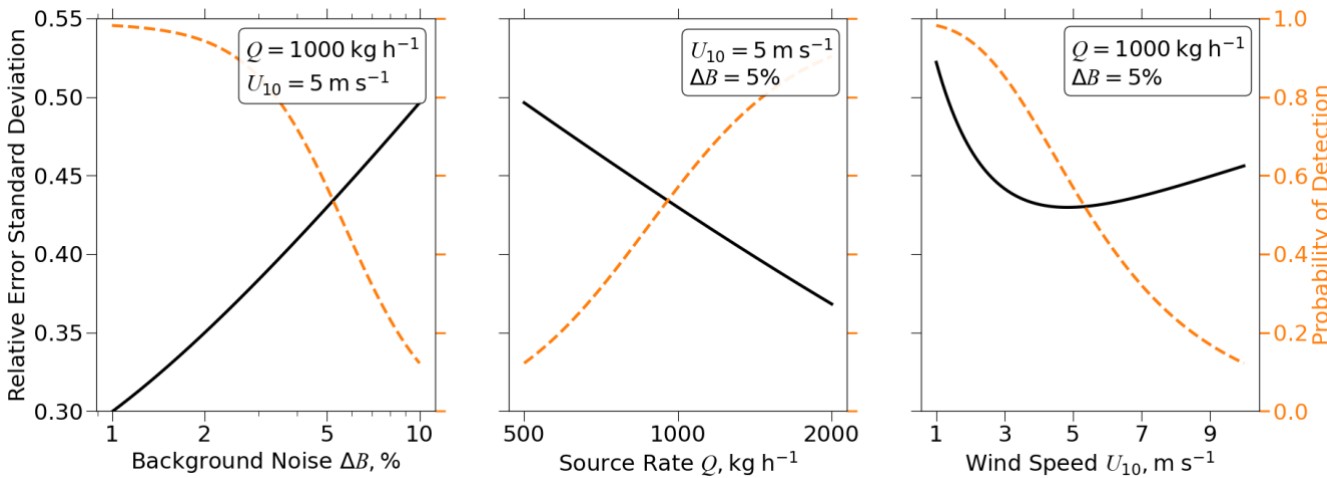



**Figure 8: Total relative error standard deviation σ$_Q$ on point source rate estimates from the U-Plume algorithm with the IME method (equation (9)) as a function of three variables: background noise $\Delta B$ (%), source rate $Q$ (kg h$^{-1}$), and 10-m wind speed $U_{10}$ (m s$^{-1}$). The dashed orange line represents the probability of point source detection calculated with equation (6).**


Adding the error variances from equations (7) and (8) in quadrature gives the total relative error variance $\sigma_Q{}^2$ on the estimated sources rates from the U-Plume algorithm including error on wind speed:

$$\sigma_Q{}^2 = \left(\max(0.1, \left(0.52 - 0.12 \log\left(O_{ps}\right)\right)\right)^2 + \left(\frac{0.23\ \sigma_U}{0.7 + 0.23\ U_{10}}\right)^2 \qquad (O_{ps} > 2) \qquad (9)$$


Fig. 8 shows the total relative error standard deviation $\sigma_Q$ as a function of background noise $\Delta B$, source rate $Q$, and 10-m wind speed, $U_{10}$. The total relative error standard deviation increases linearly with $\log(\Delta B)$, decreases linearly with $\log(Q)$, and is minimum for $U_{10} = 4$ m s$^{-1}$.





M

**Figure 9: Dependence of errors in point source estimates on wind speed. The Figure shows the relative error standard deviation on the source rate estimates inferred from U-Plume with the IME method (equation (9)) as a function of 10-m wind speed ($U_{10}$). The total relative error standard deviation $\sigma_Q$ is shown as the red line (same as Figure 8) and is decomposed into the contributions from errors in the U-Plume algorithm ($\sigma_{Q,M}$ equation (7)) and errors in wind speed ($\sigma_U$ equation (8)).**

The dependence of the total relative error standard deviation on wind speed is partitioned further in Figure 9 into the contributions from the U-Plume algorithm and wind speed errors. The error contribution from the U-Plume algorithm increases





with rising wind speed, because methane enhancements in the plume become smaller, but this levels off at high wind speeds
as the plumes are increasingly likely to be undetected instead. The error contribution from wind speed decreases with rising
wind speed. The total error is dominated by wind speed for $U_{10} < 4$ m s$^{-1}$ and by the U-Plume algorithm for $U_{10} > 4$ m s$^{-1}$. For
the range of values of $\Delta B$ and $Q$ in our dataset we find that the cross-over between the two regimes is at 2-4 m s$^{-1}$, which can
be regarded as an optimum wind speed range for plume quantification.

## 5 Conclusions

We developed the U-Plume algorithm for automated detection and quantification of point source rates from high-resolution
satellite imagery of atmospheric column concentrations. Our work was motivated by the need for fast identification of plumes
in the operational processing of the vast amounts of methane data from the rapidly expanding constellation of GHGSat and
other satellite instruments targeting methane point sources. The method can also be applied to any chemically inert plume such
as for $CO_2$.


U-Plume involves two steps. The first step is a machine learning U-Net architecture for plume detection and delineation
(masking) in noisy satellite images. The resulting plume mask is then used together with wind speed information in a second
step of source rate quantification using either a convolutional neural network (CNN) or a physically based Integrated Mass
Enhancement (IME) method. Having plume mask as well as source rate information available from U-Plume output is
important for visualization, quality control, and pinpointing the source location. The end-to-end algorithm can process 62
images per second on a single core.

We trained U-Plume with an ensemble of plume free observation images from the GHG-Sat C1 satellite instrument covering
a range of surfaces from homogeneous to highly heterogeneous and superimposing instantaneous methane plumes from large-
eddy simulations with known point source rates. Evaluation with an independent dataset shows that successful plume detection
and masking is strongly dependent on four state variables: instrument background noise $\Delta B$, source rate $Q$, 10-m wind speed
$U_{10}$, and pixel resolution $W$. We combine these variables into a new dimensionless number metric, the point source
observability $O_{ps} = Q/(WU_{10}\Delta B)$, and show that this metric can successfully predict the ability of a given plume imaging
instrument to detect the plumes and quantify source rates for given observing conditions. It can assist in determining when
automation is possible for a given target detection level. For the GHGSat-C2 instruments with background noise $\Delta B < 1\%$, we
find that U-Plume can reliably detect and quantify point sources with $Q > 400$ kg h$^{-1}$. U-Plume's value comes in the ability to
process large sets of plume data rather than pushing the limits of detectability. Human operators with knowledge of point
source location would be expected to have greater ability to detect small sources (Sherwin et al., 2022).

We find that the physically based IME method is superior to the CNN method for inferring point source rates from the plume
masks. The CNN has high bias for low source rates, and low bias for high source rates, as is typical for machine learning



methods. The superiority of the IME method would be expected considering that there is a simple physically-based linear relationship between plume mass and source rate modulated by wind speed. Although a CNN method may not require information on wind speed, such information is always available either from local measurements or from regional/global databases.

We developed an end-to-end error model for the point source rates inferred from U-Plume as a function of $O_{ps}$. We find that at low wind speeds the error is dominated by wind speed whereas at higher wind speeds the error is dominated by the U-Plume algorithm. The $U_{10}$ value where this shift occurs is typically 2-4 m s$^{-1}$. This also represents an optimal wind speed range for plume detection and quantification.

U-Plume thus offers a capable tool for fast automated processing of vast amounts of satellite imagery to detect plumes from point sources and quantify point source rates. As the ability of satellite observations to decrease background noise continues to improve, the U-Plume capability to detect and quantify points sources will correspondingly increase.

**Acknowledgments.** DJJ acknowledges funding from the NASA Carbon Monitoring System.

**Data Availability**

The model weights and both the original training data and the augmented test dataset are available through Harvard Dataverse (https://doi.org/10.7910/DVN/YFRQU4).

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
