# Peer review of "U-Plume: Automated algorithm for plume detection and source quantification by satellite point-source imagers"

_EGUsphere, 2023_

## Author Comment (AC1)

**Responses to Reviewers**

We thank the reviewers for their comments and questions. Our responses are formatted as follows:

The reviewer's comment/question (numbered) is written in black italic text.

Our responses are written in normal black text (indented).

The revised text as it appears in the manuscript is written in normal blue text (indented), with relevant changes underlined.

In responding to reviewers comments we have made changes to the manuscript primarily pertaining to more detailed description of the machine learning methods used and the use of more balanced language regarding the generalizability of this method. Additionally, the math error pointed out by reviewer 3 has been corrected and the figures and equations related to this error have been adjusted accordingly. This change does not substantively alter the content or conclusions of the paper as it was a scaling issue on a unitless number, and all related trends remain the same.

**Responses to Reviewer #1**

1. 6870 scenes were used for training. For ML, this is considered a very small data set size, and small data sets often lead to shortcomings in the trained models when compared to identical models trained on larger data sets drawn from the same distribution. How is the model performance affected when adjusting the data set size by a factor of 2 in each direction?

This is a reasonable question. These images were selected to be the largest training set from our full observation sub-setting available. We sought to avoid overfitting to the features of our background observations which may have occurred through augmentation and overuse of those images. We have added language to address this.

We then apply the random plume placement and source magnitude as described above to create 6870 simulated plume observations. This is a relatively small set for training by machine learning standards, but it is sufficient to provide a successful network which could be easily bolstered with a future collection of additional LES simulations and background images.

2. It is stated that 90% of the images were used for training and 10% for testing. In ML applications, a validation set is used to monitor for overfitting during training. Was a validation set used during training? If so, please state that along with associated information (e.g., what fraction of the data set was used for validation, what type of cross validation was used, and any early stopping criteria contingent on the validation loss). If not, please state that and justify why it was not used. If instead what the authors refer to as the testing set is actually the validation set, please correct the language accordingly,

and please introduce a testing set to statistically evaluate the generalization of the model beyond the training and validation data.

Clarifying text has been added on this question.

90% of the images are used for training and 10% are set aside for testing. We use a relatively standard configuration of the U-Net model with a modest training period and therefore do not separate a validation set for specific learning rate stopping criteria.

3. Related to the above, what loss functions are minimized over which data set (training vs. validation), and what learning rate policies and stopping criteria (if any) were used when training the ML models? How many epochs were used to train each model?

Details of the training process have been added to address these questions.

Regarding the U-Net:

We train the model for 20 epochs with a batch size of 32 images minimizing a loss function (*L*) combining binary cross-entropy (BCE, Jardon 2020) and the Jaccard score defined as:

 $L = -\ln(I(A, B)) + BCE$

(3)

**Regarding the CNN:**

The CNN structure is identical to that of the U-Net truncated at the end of the encoder path and connected to 32 and then 16 node dense layers with a single node output layer. All 3 of the added layers use a relu activation function and the model is trained using a mean squared error loss.

4. Please add details on data normalization for the inputs/outputs of the U-Net and CNN models used in this investigation. This information is necessary to include in the manuscript as it is crucial for both assessment and reproducibility.

Text has been added to clarify this point.

Because the U-Net recognizes structure, not just enhancement, the network trained on the 500-2000 kg h-1 range is still usable outside of this range as we will see. Because pixel values for methane enhancement generally fall in the -0.3-0.3 mol m-2 range we found that it is not necessary to normalize the data for training.

5. Please explicitly define the neural network architectures used in this work. The U-Net architecture is clearly defined in the source reference, so it is only necessary to state any deviations from that architecture, if any. The CNN architecture is poorly described here, providing no details on the number of layers, convolutional feature maps or kernel sizes, pooling sizes, the number of nodes in the fully-connected layers, nor activation functions. This information must be included in the manuscript for completeness.

Information regarding model architecture has been added. Additionally, a notebook containing the relevant functions for the model and training data creation has been added to the data repository

In practice, the confidence mask is generally either very close to 1 or very close to 0. Intermediate values fall off very quickly at the edge of the mask. We differ from the Ronneberger et al. structure only in that we begin with 16 convolutional layers rather than 64 and thus reach a maximum channel depth of 256 rather than 1024 at the end of the encoder path. A loss function combining binary cross entropy and Jaccard score is minimized for the training set. The code used for the model creation and loss function can be found alongside our dataset in the repository at the end of this text (https://doi.org/10.7910/DVN/YFRQU4).

The CNN method takes in a 3-channel input image containing the original image, the binary mask, and 10-m wind speed. The CNN structure is identical to that of the U-Net truncated at the end of the encoder path and connected to 32 and then 16 node dense layers with a single node output layer. All 3 of the added layers use a relu activation function and the model is trained using a mean squared error loss.

6. Related to the above, how was the CNN architecture determined for this problem? Simple grid search, Bayesian optimization, or something else? Please add details about this in the manuscript.

Details regarding the CNN (see response to question 5) have been added as well as a note on the potential for future work to improve this aspect of the method.

We retain the CNN method as an option in the U-Plume algorithm because it offers an alternative source rate estimate for verification purposes and because it could be improved in the future with a more extensive training dataset. A potential avenue for future work would be the testing of other CNN architectures and the expansion of training data to include the range of all possible source rates.

7. Lines 157-158: Which specific Intel Core i7 CPU was used for this quoted benchmark? The clock speed of i7 CPUs spans a factor of ~4 depending on architecture and TDP, ranging from under 1.1 GHz to over 4 GHz, let alone other variables such as cache amounts. Additionally, please specify whether file I/O was included in this benchmark as well as whether the data were all loaded into RAM at once or batches were loaded on the fly, as there may be significant differences in performance for these scenarios.

Details have been added for this benchmark. Clock speed is included in the original text and has been retained here.

Processing the test dataset of 687 images, preloaded from their relevant files, in 32 image batches on a single 2.6 GHz Intel Core i7 CPU takes 11 seconds, corresponding to 62 images per second.

8. This is more a general comment on the testing of the presented models. The test set solely consists of synthetic images produced in the same manner as the training data, that is, the test set does not include any real images where the predicted source rate could be compared with an existing data product. For completeness, the authors should perform some comparisons using real GHGSat-C1 scenes which contain an obvious CH4 plume that has had its source rate estimated by one or more existing methods and statistically summarize the differences between U-Plume and those methods. Ideally, many such cases should be included if feasible to better illustrate any trends in biases/deviations between this U-Plume approach and more traditional methods.

This is outside the scope of our study, but language has been generally adjusted throughout the text to reflect the focus of the study being on the relevant conclusions that can be drawn from the simulated case and the clear avenues of future work building on this study.

9. Lines 230-235: There are contradictory statements. deltaB

13. Lines 380-381: What CPU was used for this quoted benchmark? Please be specific. (See also comment #7 above.)

Specification has been added.

The algorithm can process 62 128×128 images per second on a single 2.6 GHz Intel Core i7 CPU.

14. Line 385-387: It states that "Evaluation with an independent dataset ...", but based on the described methodology earlier in the manuscript, it is misleading to describe the test set as "an independent dataset" given that it is drawn from the same distribution as the training data. It is only independent in the sense that it was not part of the training process, but the manuscript has not conclusively demonstrated that the model generalizes to independent data (real measurements). Please use more balanced language here, or perform the tests suggested above in comment #12 to better substantiate this claim.

Language has been adjusted to be more balanced and reflect the specific claim being made.

Evaluation with data unseen by the model during the training process shows that successful plume detection and masking is strongly dependent on four state variables: instrument background noise  $\Delta B$ , source rate Q, 10-m wind speed  $U_{10}$ , and pixel resolution W. 15. Figure 7: The orange line looks to be biased by the outliers at  $O_ps \sim 2$ , as the line is above the vast majority of the data for  $O_ps < 10$ . Given that  $O_ps > 30$  is omitted from the fit due to non-linearity (the error bottoms out around 10%, as mentioned), the authors may wish to consider also omitting  $O_ps

• In the conclusions, lines 383-389: The authors state that "the point source observability metric can successfully predict the ability of a given plume imaging instrument to detect the plumes and quantify source rates for given observing conditions", but no real demonstrations have been made apart from the simulations with GHGSat-C1.

We have adjusted the text to reflect the limitations of the claims to the instrument used for the study and added more balanced language as to the future application to other instruments.

the point source observability  $O_{ps} = Q/(WU_{10}\Delta B)$ , and show that this metric can successfully predict the ability of a the GHGSat-C1 imager to detect the plumes and quantify source rates for given observing conditions. As it is dependent on the fundamental state variables of the enhancement produced by a methane point source, we expect that the point source observability will be a valuable metric for evaluating detectability and quantification accuracy of other point source imagers after appropriate tuning.

Related to the previous point, all the analysis has been done with simulations and no real plume. I think the manuscript would be much more complete if its effectiveness were tested with real plumes, either with GHGSat-C1 detections in the simplest case or by applying it to images from other point source instruments.

This is outside the scope of our study, but language has been generally adjusted throughout the text to reflect the focus of the study being on the relevant conclusions that can be drawn from the simulated case and the clear avenues of future work building on this study.

In section 4.3. Error analysis: I miss some discussion against the results obtained by Gorroño et al., 2023 (https://amt.copernicus.org/articles/16/89/2023/amt-16-89-2023.html), who did this same error analysis applied to Sentinel-2.

While similar in goals and use of IME we do not believe the analyses are sufficiently similar to warrant direct comparison in the paper as our error analysis largely centers around the newly defined Ops value.

**Minor comments:**

1. I would say that the correct term would be "point source emission observability" instead of "point source observability". The authors repeat the same error throughout the text; for example, in the abstract, line 10, detection and quantification of point sources = point source emissions. Please check the rest of the cases.

The point source observability is used in both the context of determining the ability for an emission to be detected at all (fig. 4,5) as well as a predictor of source rate estimation accuracy (fig 7,8) and we therefore respectfully advocate for keeping with our original naming convention.

2. Line 28: "spatial resolution finer than 50 m", EMIT has 60 m resolution. Even if it is not a satellite but a sensor on the ISS, given its contribution to the methane point source emission detections, I would suggest including it and changing 50 m for 60 m.

The text has been altered to reflect this observation.

A number of satellite instruments can now detect and image methane column plumes with spatial resolution finer than 60 m by solar backscatter in the shortwave infrared (SWIR), enabling quantification of large point sources from individual facilities (Jacob et al., 2022).

3. The references in line 44 Frankenberg et al., 2016 and Cusworth et al., 2021 are references from airborne campaigns. I would recommend putting more proper references from satellite/space sensor surveys, e.g., Thorpe et al., 2023 https://www.science.org/doi/10.1126/sciadv.adh2391, Irakulis-Loitxate et al., 2021 https://www.science.org/doi/epdf/10.1126/sciadv.abf4507, Ehret et al., 2022, https://pubs.acs.org/doi/10.1021/acs.est.1c08575

The original references have been retained for completeness, but the additional literature references have been added.

but is difficult to implement for methane plumes that are typically smaller and intermittent (Frankenberg et al., 2016; Cusworth et al., 2021, Irakulis-Loitxate et al., 2021, Ehret et al., 2022, Thorpe et al., 2023).

4. Line 184: comma after "In practice" instead of a dot.

Correction has been made.

In practice, the confidence mask is generally either very close to 1 or very close to 0. Intermediate values fall off very quickly at the edge of the mask.

**5. Equation (3) and lines 205-206: I assume that this Ueff calibration is specific for GHGSat-C1 detections; please clarify whether this is the case or not.**

This is correct. Text has been added for clarification.

The plume length scale L is defined as the square root of the plume mask area. The effective wind speed is fitted to the 15-minute averaged wind speed  $U_{10}$  at 10-m altitude using the training data set where the relationship between Q, IME, and L is known for a given  $U_{10}$ . This yields

$$U_{eff} = 0.7 + 0.23 \, U_{10} \tag{5}$$

with  $U_{eff}$  and  $U_{10}$  in units of m s-1. The intercept of 0.7 m s-1 arising from the fit can be physically interpreted as a minimum turbulent diffusion for the plume at low wind speeds. This calibration is specific to the GHGSat-C1 instrument and should be recalculated when applying U-Plume to other platforms.

**6. Line 233: "...for high background noise ( $\Delta B < 20\%$ )" instead of <5%?**

There was a typo as we mean to divide into regimes of >5% and

The U-net network is described in the source reference, but there is no description of the architecture of the CNN – please mention number of layers, size of layers (e.g. number of nodes, pooling sizes, kernel sizes), activation functions, loss function etc.

We now clarify the manners in which our architecture differs from the cited architecture.

In practice, the confidence mask is generally either very close to 1 or very close to 0. Intermediate values fall off very quickly at the edge of the mask. We differ from the Ronneberger et al. structure only in that we begin with 16 convolutional layers rather than 64 and thus reach a maximum channel depth of 256 rather than 1024 at the end of the encoder path. A loss function combining binary cross

entropy and Jaccard score is minimized for the training set. The code used for the model creation and loss function can be found alongside our dataset in the repository at the end of this text (https://doi.org/10.7910/DVN/YFRQU4)

Include details on training for both nets, e.g. learning rates, epochs, batch sizes...

Information regarding model architecture has been added. Additionally, a notebook containing the relevant functions for the model and training data creation has been added to the data repository

In practice, the confidence mask is generally either very close to 1 or very close to 0. Intermediate values fall off very quickly at the edge of the mask. We differ from the Ronneberger et al. structure only in that we begin with 16 convolutional layers rather than 64 and thus reach a maximum channel depth of 256 rather than 1024 at the end of the encoder path. A loss function combining binary cross entropy and Jaccard score is minimized for the training set. The code used for the model creation and loss function can be found alongside our dataset in the repository at the end of this text (https://doi.org/10.7910/DVN/YFRQU4).

The CNN method takes in a 3-channel input image containing the original image, the binary mask, and 10-m wind speed. The CNN structure is identical to that of the U-Net truncated at the end of the encoder path and connected to 32 and then 16 node dense layers with a single node output layer. All 3 of the added layers use a relu activation function and the model is trained using a mean squared error loss.

**The CNN is said to not perform well outside of the training. Comment/demonstrate how the performance would improve if trained on these source rates.**

Language has been added to this affect.

We retain the CNN method as an option in the U-Plume algorithm because it offers an alternative source rate estimate for verification purposes and because it could be improved in the future with a more extensive training dataset. A potential avenue for future work would be the testing of other CNN architectures and the expansion of training data to include the range of all possible source rates.

It is not clear if the CNN is trained independently (i.e. using true masked footprints and their corresponding source rate) or within the pipeline (i.e. using masked footprints outputted by the U-net, and their source rate). Clarify, and comment on the potential impact of this on the results. If trained independently, evaluate the differences in performance using perfectly masked footprints vs the output of the U-net.

This has been clarified, we are training using the U-Net masks from successful masking cases.

We train the model for 10 epochs with a batch size of 32 images and use U-Net generated masks for the mask input channel (limited to successfully masked examples).

3. Algebra error

For the proposed equations to work, Ops seems to be off by a factor of 100. The equations only hold if Ops = 100Q/UW deltaB rather than the current equation 4 Ops = Q/UW deltaB. Demonstration by example:

From the third panel in figure 8, probability of detection is around 0.2 for wind speed 9, Q 1000kgh and deltaB 5%. Calculating Ops using these and W=25, and accounting for the change in units in Q from kgh to kgs, Ops = (Q/3600)/UWdeltaB = (1000/3600)/(9\*25\*0.05)=0.0247, which is below the lower boundary of Ops as defined in Eq 6.

Instead, working backwards using Eq 6: for a detection probability of 0.2, Ops = 2.47, which is off by a factor of a 100 from the Ops calculated using the original formula and parameters.

Great catch! There was an error in the code resulting in this issue that has now been corrected. We are so grateful for the reviewer catching this! Figures 4 and 7 have been adjusted to reflect the correct values for Ops.

Detection probability is 10% for  $O_{ps} = \underline{0.02}$ , 50% for  $O_{ps} = \underline{0.04}$ , and 90% for  $O_{ps} = \underline{0.08}$ .

... and we have added a sentence in the acknowledgments,

'We thank an anonymous reviewer for identifying a coding error in the original version.'

**4. Results**

The results are shown for selected value combinations of Q, deltaB and U. How are these value combinations chosen, and how does performance change outside of these combinations?

Clarifying text has been added.

Fig. 8 shows the total relative error standard deviation  $\sigma_Q$  as a function of background noise  $\Delta B$ , source rate Q, and 10-m wind speed,  $U_{10}$ . We attempt to represent the range of common observing conditions, but equation 10 can be used to calculate the values at any desired combination within the limits of the composing equations.

Claims on the effectiveness of this method are not backed up sufficiently, e.g. line 390 "For the GHGSat-C2 instruments with background noise deltaB < 1%, we find that Uplume can reliably detect and quantify point sources with Q>400kgh". This is not demonstrated in the manuscript, as the training and testing is conducted only on synthetic data. Please evaluate on real GHGSat data to support this claim, or clarify. Similarly, it is not demonstrated that "this method can be applied to any chemically inert plume such as CO2" (line 373)

Language has been adjusted to be more balanced.

It can assist in determining when automation is possible for a given target detection level. For the GHGSat-C2 instruments with background noise  $\Delta B < 1\%$ , we expect that U-Plume could reliably detect and quantify point sources with  $Q > 400 \text{ kg h}^{-1}$ .

The method can also be applied to any chemically inert plume such as for CO2 with appropriate instrument specific training due to the inherent similarity in the physics governing plume formation and transport.

**Technical Corrections**

*Line 184: Typo "In practice. the confidence mask"*

Correction has been made.

In practice, the confidence mask is generally either very close to 1 or very close to 0. Intermediate values fall off very quickly at the edge of the mask.

*Eq 4: Clarify that in this work, W corresponds to 25m.*

Clarification has been added.

Observability is a function of source rate (Q, kg s-1), wind speed (U, m s-1), instrument pixel resolution (W, m) which is 25 m for our case, and scenedependent background noise ( $\Delta B$ , kg m-2, as defined by equation (1)).

*Line 233: Typo "… for high background noise (deltaB<5%)"*

This has been corrected.

These failures occur for high background noise ( $\Delta B \ge 5\%$ ) and lower emission rates. On the other hand, Jaccard scores generally exceed 0.5 for low background noise ( $\Delta B < 5\%$ ).

**Line 252: Missing full stop after Ops**

Added

Here we use the 10-m wind speed  $(U_{10})$  in the calculation of  $O_{ps.}$

Line 262: Typo "Detection probability is 10% for Ops = 0.2". Ops lower bound is defined as 1.4 in eq 6, so this is likely a typo, meaning to say for Ops=2.

This has been corrected to be in line with the corrected Ops values as discussed under main comment 4.

Detection probability is 10% for  $O_{ps} = \underline{0.02}$ , 50% for  $O_{ps} = \underline{0.04}$ , and 90% for  $O_{ps} = \underline{0.08}$ .

---

## Author Comment (AC2)

**Responses to Reviewers**

We thank the reviewers for their comments and questions. Our responses are formatted as follows:

*The reviewer's comment/question (numbered) is written in black italic text.*

      Our responses are written in normal black text (indented).

      The revised text as it appears in the manuscript is written in normal blue text (indented), with relevant changes underlined.

In responding to reviewers comments we have made changes to the manuscript primarily pertaining to more detailed description of the machine learning methods used and the use of more balanced language regarding the generalizability of this method. Additionally, the math error pointed out by reviewer 3 has been corrected and the figures and equations related to this error have been adjusted accordingly. This change does not substantively alter the content or conclusions of the paper as it was a scaling issue on a unitless number, and all related trends remain the same.

**Responses to Reviewer #1**

1. *6870 scenes were used for training. For ML, this is considered a very small data set size, and small data sets often lead to shortcomings in the trained models when compared to identical models trained on larger data sets drawn from the same distribution. How is the model performance affected when adjusting the data set size by a factor of 2 in each direction?*

      This is a reasonable question. These images were selected to be the largest training set from our full observation sub-setting available. We sought to avoid overfitting to the features of our background observations which may have occurred through augmentation and overuse of those images. We have added language to address this.

      We then apply the random plume placement and source magnitude as described above to create 6870 simulated plume observations. This is a relatively small set for training by machine learning standards, but it is sufficient to provide a successful network which could be easily bolstered with a future collection of additional LES simulations and background images.

2. *It is stated that 90% of the images were used for training and 10% for testing. In ML applications, a validation set is used to monitor for overfitting during training. Was a validation set used during training? If so, please state that along with associated information (e.g., what fraction of the data set was used for validation, what type of cross validation was used, and any early stopping criteria contingent on the validation loss). If not, please state that and justify why it was not used. If instead what the authors refer to as the testing set is actually the validation set, please correct the language accordingly,*

*and please introduce a testing set to statistically evaluate the generalization of the model beyond the training and validation data.*

Clarifying text has been added on this question.

90% of the images are used for training and 10% are set aside for testing. We use a relatively standard configuration of the U-Net model with a modest training period and therefore do not separate a validation set for specific learning rate stopping criteria.

3. *Related to the above, what loss functions are minimized over which data set (training vs. validation), and what learning rate policies and stopping criteria (if any) were used when training the ML models? How many epochs were used to train each model?*

Details of the training process have been added to address these questions.

Regarding the U-Net:

We train the model for 20 epochs with a batch size of 32 images minimizing a loss function ($L$) combining binary cross-entropy (BCE, Jardon 2020) and the Jaccard score defined as:

$$L = -\ln\big(J(A,B)\big) + \text{BCE} \tag{3}$$

Regarding the CNN:

The CNN structure is identical to that of the U-Net truncated at the end of the encoder path and connected to 32 and then 16 node dense layers with a single node output layer. All 3 of the added layers use a relu activation function and the model is trained using a mean squared error loss.

4. *Please add details on data normalization for the inputs/outputs of the U-Net and CNN models used in this investigation. This information is necessary to include in the manuscript as it is crucial for both assessment and reproducibility.*

Text has been added to clarify this point.

Because the U-Net recognizes structure, not just enhancement, the network trained on the 500-2000 kg h$^{-1}$ range is still usable outside of this range as we will see. Because pixel values for methane enhancement generally fall in the -0.3-0.3 mol m$^{-2}$ range we found that it is not necessary to normalize the data for training.

5. *Please explicitly define the neural network architectures used in this work. The U-Net architecture is clearly defined in the source reference, so it is only necessary to state any deviations from that architecture, if any. The CNN architecture is poorly described here, providing no details on the number of layers, convolutional feature maps or kernel sizes, pooling sizes, the number of nodes in the fully-connected layers, nor activation functions. This information must be included in the manuscript for completeness.*

Information regarding model architecture has been added. Additionally, a notebook containing the relevant functions for the model and training data creation has been added to the data repository

In practice, the confidence mask is generally either very close to 1 or very close to 0. Intermediate values fall off very quickly at the edge of the mask. We differ from the Ronneberger et al. structure only in that we begin with 16 convolutional layers rather than 64 and thus reach a maximum channel depth of 256 rather than 1024 at the end of the encoder path. A loss function combining binary cross entropy and Jaccard score is minimized for the training set. The code used for the model creation and loss function can be found alongside our dataset in the repository at the end of this text (https://doi.org/10.7910/DVN/YFRQU4).

The CNN method takes in a 3-channel input image containing the original image, the binary mask, and 10-m wind speed. The CNN structure is identical to that of the U-Net truncated at the end of the encoder path and connected to 32 and then 16 node dense layers with a single node output layer. All 3 of the added layers use a relu activation function and the model is trained using a mean squared error loss.

6. *Related to the above, how was the CNN architecture determined for this problem? Simple grid search, Bayesian optimization, or something else? Please add details about this in the manuscript.*

Details regarding the CNN (see response to question 5) have been added as well as a note on the potential for future work to improve this aspect of the method.

We retain the CNN method as an option in the U-Plume algorithm because it offers an alternative source rate estimate for verification purposes and because it could be improved in the future with a more extensive training dataset. A potential avenue for future work would be the testing of other CNN architectures and the expansion of training data to include the range of all possible source rates.

7. *Lines 157-158: Which specific Intel Core i7 CPU was used for this quoted benchmark? The clock speed of i7 CPUs spans a factor of ~4 depending on architecture and TDP, ranging from under 1.1 GHz to over 4 GHz, let alone other variables such as cache amounts. Additionally, please specify whether file I/O was included in this*

*benchmark as well as whether the data were all loaded into RAM at once or batches were loaded on the fly, as there may be significant differences in performance for these scenarios.*

Details have been added for this benchmark. Clock speed is included in the original text and has been retained here.

Processing the test dataset of 687 images, preloaded from their relevant files, in 32 image batches on a single 2.6 GHz Intel Core i7 CPU takes 11 seconds, corresponding to 62 images per second.

8. *This is more a general comment on the testing of the presented models. The test set solely consists of synthetic images produced in the same manner as the training data, that is, the test set does not include any real images where the predicted source rate could be compared with an existing data product. For completeness, the authors should perform some comparisons using real GHGSat-C1 scenes which contain an obvious CH4 plume that has had its source rate estimated by one or more existing methods and statistically summarize the differences between U-Plume and those methods. Ideally, many such cases should be included if feasible to better illustrate any trends in biases/deviations between this U-Plume approach and more traditional methods.*

This is outside the scope of our study, but language has been generally adjusted throughout the text to reflect the focus of the study being on the relevant conclusions that can be drawn from the simulated case and the clear avenues of future work building on this study.

9. *Lines 230-235: There are contradictory statements. deltaB < 5% is described as both "high background noise" and "low background noise", but only one of these can be true. Please correct this typo so that readers may understand what the authors consider to be low/high background noise.*

This typo has been clarified to explain that we consider 5% to be the transition point from our low to high background noise regimes.

These failures occur for high background noise ($\Delta B \geq 5\%$ ) and lower emission rates. On the other hand, Jaccard scores generally exceed 0.5 for low background noise ($\Delta B < 5\%$).

10. *Lines 239-242: Can you comment more on these false positives? How are they distributed with respect to the variables considered in this study? Were any other approaches considered to address them? Are the estimated source rates from these false positives generally small and could be filtered that way rather than based on number of pixels in the mask?*
*Using the 5-pixel-mask filtering loses 1.6% of the true positive detections; can you comment more on this? Are these generally small source rates at low wind speeds, cases*

with high background noise, or are they more uniformly distributed throughout the domain of interest?

Additional discussion of the false positives has been added to address these questions.

Applying a minimum 5-pixel mask size brings the false positive rate down to 6% while losing only 1.6% of the true positive detections in the test data set and removing 92% of the zero overlap masks. False positives occur more comonly in scenes with high $\Delta B$ values and particularly in scenes with complex topography. The mask sizes for these false positives are generally small, as is apparent in the mask size filtering effectiveness, and their corresponding estimated source rates are therefore also generally small.

11. *Line 262: It says that "Detection probability is 10% for O_ps = 0.2".  In Figure 4, O_ps has a minimum value of 0.5, at which it has a detection probability of 0%.  O_ps = 2 looks to be closer to the 10% detection probability mentioned.  Please correct this typo.*

We have corrected these values as well as making a math correction suggested by Reviewer 3 and the text is now in line with the results shown in the figure.

Detection probability is 10% for $O_{ps} = 0.02$, 50% for $O_{ps} = 0.04$, and 90% for $O_{ps} = 0.08$.

12. *Lines 291-298: It is mentioned that the CNN method is biased towards the mean, that is, it overestimates small sources and underestimates large sources.  What is the distribution of source rates in the training set used in this investigation?  If that distribution is biased towards the mean, then that could also explain the CNN's reported behavior.  If that is the case, the authors should either address this bias in the data set to improve model performance at the extrema (whether in the data set itself, or in the loss function used to train the model) or use more balanced language when discussing this limitation.*
*This bias towards the mean can also be a consequence of how the CNN was trained (though the manuscript lacks sufficient detail on how these models were trained to determine the likelihood of this being the case - see comments above).*
*While the CNN is likely to still perform poorly when extrapolating regardless of data set, the authors have not conclusively ruled out bias in the training data set or particular training methodology as the reason for the CNN's worse performance vs. the IME method over the domain that the CNN was trained on.  Furthermore, it should be mentioned that expanding the training data set down to 100 kg/h source rates would likely enable the CNN to more accurately recover those scenarios.*

Clarifying information has been added regarding the CNN structure:

The CNN method takes in a 3-channel input image containing the original image, the binary mask, and 10-m wind speed. The CNN structure is identical to that of

the U-Net truncated at the end of the encoder path and connected to 32 and then 16 node dense layers with a single node output layer. All 3 of the added layers use a relu activation function and the model is trained using a mean squared error loss. We train the model for 10 epochs with a batch size of 32 images and use U-Net generated masks for the mask input channel (limited to successfully masked examples).

We add more balanced language when discussing the results from the CNN estimations.

We retain the CNN method as an option in the U-Plume algorithm because it offers an alternative source rate estimate for verification purposes and because it could be improved in the future with a more extensive training dataset. A potential avenue for future work would be the testing of other CNN architectures and the expansion of training data to include the range of all possible source rates.

13. *Lines 380-381: What CPU was used for this quoted benchmark? Please be specific. (See also comment #7 above.)*

Specification has been added.

The algorithm can process 62 128×128 images per second on a single 2.6 GHz Intel Core i7 CPU.

14. *Line 385-387: It states that "Evaluation with an independent dataset ...", but based on the described methodology earlier in the manuscript, it is misleading to describe the test set as "an independent dataset" given that it is drawn from the same distribution as the training data. It is only independent in the sense that it was not part of the training process, but the manuscript has not conclusively demonstrated that the model generalizes to independent data (real measurements). Please use more balanced language here, or perform the tests suggested above in comment #12 to better substantiate this claim.*

Language has been adjusted to be more balanced and reflect the specific claim being made.

Evaluation with data unseen by the model during the training process shows that successful plume detection and masking is strongly dependent on four state variables: instrument background noise $\Delta B$, source rate $Q$, 10-m wind speed $U_{10}$, and pixel resolution $W$.

15. *Figure 7: The orange line looks to be biased by the outliers at O_ps ~ 2, as the line is above the vast majority of the data for O_ps < 10. Given that O_ps > 30 is omitted from the fit due to non-linearity (the error bottoms out around 10%, as mentioned), the authors may wish to consider also omitting O_ps < 3 from the fit due to non-linearity.*

> This is a good observation and suggestion. We have implemented this change to the regression, and it is now reflected in figure 7 and the accompanying text and equation.

16. *Figure 8: In the left and middle panels, some of the plotted lines are covered by the legend. Please relocate the legend in these panels to avoid this behavior. In the left panel, it looks like it could be placed in the center-left, while the middle panel could relocate the legend to the top-left or bottom-right of the plot.*

> The suggested alteration to the figure legend position has been made.

**Responses to Reviewer #2**

1. *The authors say that U-plume is applicable to all point source imagers, but the performance analysis is based entirely on GHGat-C1 images, which are not publicly available. I miss an analysis or test of the algorithm's effectiveness applied to publicly available data from other missions more commonly used by the scientific community, e.g., Sentinel-2 or PRISMA. If this is out of the scope of the study, I would suggest making the manuscript more explicit that the results presented here are applicable to GHGSat-C1 and escaping from the narrative that these results are to be expected with other missions. This would apply, for example:*

   - *In the abstract, line 20: "U-Plume successfully detects and masks plumes from sources as small as 100 kg h-1", please add "in GHGSat-C1 images"*

     > We have made the suggested alteration.

     > U-Plume successfully detects and masks plumes from sources as small as 100 kg h$^{-1}$ in GHGSat-C1 images over surfaces with low background noise and succeeds for larger point sources over surfaces with substantial background noise.

   - *First paragraph of the introduction, line 32: how do you know you are addressing the need described in the paragraph if you have only applied it to the GHSat data?*

     > Clarifying text has been added throughout to specify the results being for GHGSat-C1 data and the generalizable portion of the work being rooted in the model and training approach

- *Lines 85-87: how do you know it works if you have not done any other tests? If you want to escape from testing it on other point source imagers, I would propose changing the last sentence to "but the architecture and training process presented here are potentially applicable to other point source imagers for other species."*

    We have made the suggested alteration.

    We demonstrate U-Plume on the GHGSat-C1 instrument for methane, but the architecture and training process presented here are potentially applicable to any satellite point source imagers for any species.

- *In the conclusions, lines 383-389: The authors state that "the point source observability metric can successfully predict the ability of a given plume imaging instrument to detect the plumes and quantify source rates for given observing conditions", but no real demonstrations have been made apart from the simulations with GHGSat-C1.*

    We have adjusted the text to reflect the limitations of the claims to the instrument used for the study and added more balanced language as to the future application to other instruments.

    the point source observability $O_{ps} = Q/(WU_{10}\Delta B)$, and show that this metric can successfully predict the ability of a the GHGSat-C1 imager to detect the plumes and quantify source rates for given observing conditions. As it is dependent on the fundamental state variables of the enhancement produced by a methane point source, we expect that the point source observability will be a valuable metric for evaluating detectability and quantification accuracy of other point source imagers after appropriate tuning.

*Related to the previous point, all the analysis has been done with simulations and no real plume. I think the manuscript would be much more complete if its effectiveness were tested with real plumes, either with GHGSat-C1 detections in the simplest case or by applying it to images from other point source instruments.*

    This is outside the scope of our study, but language has been generally adjusted throughout the text to reflect the focus of the study being on the relevant conclusions that can be drawn from the simulated case and the clear avenues of future work building on this study.

*In section 4.3. Error analysis: I miss some discussion against the results obtained by Gorroño et al., 2023 (https://amt.copernicus.org/articles/16/89/2023/amt-16-89-2023.html), who did this same error analysis applied to Sentinel-2.*

While similar in goals and use of IME we do not believe the analyses are sufficiently similar to warrant direct comparison in the paper as our error analysis largely centers around the newly defined Ops value.

*Minor comments:*

1. *I would say that the correct term would be "point source emission observability" instead of "point source observability". The authors repeat the same error throughout the text; for example, in the abstract, line 10, detection and quantification of point sources = point source emissions. Please check the rest of the cases.*

    The point source observability is used in both the context of determining the ability for an emission to be detected at all (fig. 4,5) as well as a predictor of source rate estimation accuracy (fig 7,8) and we therefore respectfully advocate for keeping with our original naming convention.

2. *Line 28: "spatial resolution finer than 50 m", EMIT has 60 m resolution. Even if it is not a satellite but a sensor on the ISS, given its contribution to the methane point source emission detections, I would suggest including it and changing 50 m for 60 m.*

    The text has been altered to reflect this observation.

    A number of satellite instruments can now detect and image methane column plumes with spatial resolution finer than 60 m by solar backscatter in the shortwave infrared (SWIR), enabling quantification of large point sources from individual facilities (Jacob et al., 2022).

3. *The references in line 44 Frankenberg et al., 2016 and Cusworth et al., 2021 are references from airborne campaigns. I would recommend putting more proper references from satellite/space sensor surveys, e.g., Thorpe et al., 2023 https://www.science.org/doi/10.1126/sciadv.adh2391, Irakulis-Loitxate et al., 2021 https://www.science.org/doi/epdf/10.1126/sciadv.abf4507, Ehret et al., 2022, https://pubs.acs.org/doi/10.1021/acs.est.1c08575*

    The original references have been retained for completeness, but the additional literature references have been added.

    but is difficult to implement for methane plumes that are typically smaller and intermittent (Frankenberg et al., 2016; Cusworth et al., 2021, Irakulis-Loitxate et al., 2021, Ehret et al., 2022, Thorpe et al., 2023).

4. *Line 184: comma after "In practice" instead of a dot.*

    Correction has been made.

In practice, the confidence mask is generally either very close to 1 or very close to 0. Intermediate values fall off very quickly at the edge of the mask.

5. *Equation (3) and lines 205-206: I assume that this Ueff calibration is specific for GHGSat-C1 detections; please clarify whether this is the case or not.*

This is correct. Text has been added for clarification.

The plume length scale $L$ is defined as the square root of the plume mask area. The effective wind speed is fitted to the 15-minute averaged wind speed $U_{10}$ at 10-m altitude using the training data set where the relationship between $Q$, IME, and $L$ is known for a given $U_{10}$. This yields

$$U_{eff} = 0.7 + 0.23 \ U_{10} \tag{5}$$

with $U_{eff}$ and $U_{10}$ in units of m s$^{-1}$. The intercept of 0.7 m s$^{-1}$ arising from the fit can be physically interpreted as a minimum turbulent diffusion for the plume at low wind speeds. This calibration is specific to the GHGSat-C1 instrument and should be recalculated when applying U-Plume to other platforms.

6. *Line 233: "...for high background noise (ΔB < **20%**)" instead of <5%?*

There was a typo as we mean to divide into regimes of >5% and <5%. This error has been corrected.

These failures occur for high background noise ($\Delta B \geq 5\%$) and lower emission rates. On the other hand, Jaccard scores generally exceed 0.5 for low background noise ($\Delta B < 5\%$).

**Responses to Reviewer #3**

1. *Data generation reproducibility – there is little description of how the dataset is generated. These should be included in the manuscript or as an appendix:*

*Line 103: "We conduct 4 simulations for different meteorological conditions". What are the conditions?*

As we are using the same LES simulations as a previous study the detailed description is left to the reference, but we have added text to clarify this and properly point the reader

This is the same ensemble as used by Varon et al (2021) where the specific details of the simulations are provided.

*Line 120: "We use 28 plume-free observations corresponding to a variety of surface types": What are the locations for these observations? What is the distribution of background noises across the 28 observations?*

Added some text regarding the range of noise values across the sub-setted observations.

This filtering leaves us with a total of 6870 scenes with a median $\Delta B$ of 8% and interquartiles of 5% and 12%.

*Line 152: "90% of the images are used for training and 10% are set aside for testing". Is the split random, and could this affect the quality of the model? Is a validation set defined for tuning, and if not why?*

Clarifying text has been added on this question.

90% of the images are used for training and 10% are set aside for testing. We use a relatively standard configuration of the U-Net model with a modest training period and therefore do not separate a validation set for specific learning rate stopping criteria.

2. *ML model reproducibility – there is little description of the architecture of the model and its process:*

   *Data normalisation and pre-processing: how was the image data and the wind-speed modified before being inputted to the neural network?*

   We have clarified that the data is not normalized.

   Because the U-Net recognizes structure, not just enhancement, the network trained on the 500-2000 kg h$^{-1}$ range is still usable outside of this range as we will see. Because pixel values for methane enhancement generally fall in the -0.3-0.3 mol m$^{-2}$ range we found that it is not necessary to normalize the data for training.

   *The U-net network is described in the source reference, but there is no description of the architecture of the CNN – please mention number of layers, size of layers (e.g. number of nodes, pooling sizes, kernel sizes), activation functions, loss function etc.*

   We now clarify the manners in which our architecture differs from the cited architecture.

   In practice, the confidence mask is generally either very close to 1 or very close to 0. Intermediate values fall off very quickly at the edge of the mask. We differ from the Ronneberger et al. structure only in that we begin with 16 convolutional layers rather than 64 and thus reach a maximum channel depth of 256 rather than 1024 at the end of the encoder path. A loss function combining binary cross

entropy and Jaccard score is minimized for the training set. The code used for the model creation and loss function can be found alongside our dataset in the repository at the end of this text (https://doi.org/10.7910/DVN/YFRQU4)

*Include details on training for both nets, e.g. learning rates, epochs, batch sizes…*

Information regarding model architecture has been added. Additionally, a notebook containing the relevant functions for the model and training data creation has been added to the data repository

In practice, the confidence mask is generally either very close to 1 or very close to 0. Intermediate values fall off very quickly at the edge of the mask. We differ from the Ronneberger et al. structure only in that we begin with 16 convolutional layers rather than 64 and thus reach a maximum channel depth of 256 rather than 1024 at the end of the encoder path. A loss function combining binary cross entropy and Jaccard score is minimized for the training set. The code used for the model creation and loss function can be found alongside our dataset in the repository at the end of this text (https://doi.org/10.7910/DVN/YFRQU4).

The CNN method takes in a 3-channel input image containing the original image, the binary mask, and 10-m wind speed. The CNN structure is identical to that of the U-Net truncated at the end of the encoder path and connected to 32 and then 16 node dense layers with a single node output layer. All 3 of the added layers use a relu activation function and the model is trained using a mean squared error loss.

*The CNN is said to not perform well outside of the training. Comment/demonstrate how the performance would improve if trained on these source rates.*

Language has been added to this affect.

We retain the CNN method as an option in the U-Plume algorithm because it offers an alternative source rate estimate for verification purposes and because it could be improved in the future with a more extensive training dataset. A potential avenue for future work would be the testing of other CNN architectures and the expansion of training data to include the range of all possible source rates.

*It is not clear if the CNN is trained independently (i.e. using true masked footprints and their corresponding source rate) or within the pipeline (i.e. using masked footprints outputted by the U-net, and their source rate). Clarify, and comment on the potential impact of this on the results. If trained independently, evaluate the differences in performance using perfectly masked footprints vs the output of the U-net.*

This has been clarified, we are training using the U-Net masks from successful masking cases.

We train the model for 10 epochs with a batch size of 32 images and use U-Net generated masks for the mask input channel (limited to successfully masked examples).

3. *Algebra error*

*For the proposed equations to work, Ops seems to be off by a factor of 100. The equations only hold if Ops = 100Q/UWdeltaB rather than the current equation 4 Ops = Q/UWdeltaB. Demonstration by example:*

*From the third panel in figure 8, probability of detection is around 0.2 for wind speed 9, Q 1000kgh and deltaB 5%. Calculating Ops using these and W=25, and accounting for the change in units in Q from kgh to kgs, Ops = (Q/3600)/UWdeltaB = (1000/3600)/(9\*25\*0.05)=0.0247, which is below the lower boundary of Ops as defined in Eq 6.*

*Instead, working backwards using Eq 6: for a detection probability of 0.2, Ops = 2.47, which is off by a factor of a 100 from the Ops calculated using the original formula and parameters.*

Great catch! There was an error in the code resulting in this issue that has now been corrected. We are so grateful for the reviewer catching this! Figures 4 and 7 have been adjusted to reflect the correct values for Ops.

Detection probability is 10% for $O_{ps}$ = 0.02, 50% for $O_{ps}$ = 0.04, and 90% for $O_{ps}$ = 0.08.

…and we have added a sentence in the acknowledgments,

'We thank an anonymous reviewer for identifying a coding error in the original version.'

4. *Results*

*The results are shown for selected value combinations of Q, deltaB and U. How are these value combinations chosen, and how does performance change outside of these combinations?*

Clarifying text has been added.

Fig. 8 shows the total relative error standard deviation $\sigma_Q$ as a function of background noise $\Delta B$, source rate $Q$, and 10-m wind speed, $U_{10}$. We attempt to represent the range of common observing conditions, but equation 10 can be used to calculate the values at any desired combination within the limits of the composing equations.

*Claims on the effectiveness of this method are not backed up sufficiently, e.g. line 390 "For the GHGSat-C2 instruments with background noise deltaB < 1%, we find that U-plume can reliably detect and quantify point sources with Q>400kgh". This is not demonstrated in the manuscript, as the training and testing is conducted only on synthetic data. Please evaluate on real GHGSat data to support this claim, or clarify. Similarly, it is not demonstrated that "this method can be applied to any chemically inert plume such as CO2" (line 373)*

Language has been adjusted to be more balanced.

It can assist in determining when automation is possible for a given target detection level. For the GHGSat-C2 instruments with background noise $\Delta B < 1\%$, we expect that U-Plume could reliably detect and quantify point sources with $Q > 400$ kg h$^{-1}$.

The method can also be applied to any chemically inert plume such as for $CO_2$ with appropriate instrument specific training due to the inherent similarity in the physics governing plume formation and transport.

**Technical Corrections**

*Line 184: Typo "In practice. the confidence mask"*

Correction has been made.

In practice, the confidence mask is generally either very close to 1 or very close to 0. Intermediate values fall off very quickly at the edge of the mask.

*Eq 4: Clarify that in this work, W corresponds to 25m.*

Clarification has been added.

Observability is a function of source rate ($Q$, kg s$^{-1}$), wind speed ($U$, m s$^{-1}$), instrument pixel resolution ($W$, m) which is 25 m for our case, and scene-dependent background noise ($\Delta B$, kg m$^{-2}$, as defined by equation (1)).

*Line 233: Typo "… for high background noise (deltaB<5%)"*

This has been corrected.

These failures occur for high background noise ($\Delta B \geq 5\%$) and lower emission rates. On the other hand, Jaccard scores generally exceed 0.5 for low background noise ($\Delta B < 5\%$).

*Line 252: Missing full stop after Ops*

Added

Here we use the 10-m wind speed ($U_{10}$) in the calculation of $O_{ps}$.

*Line 262: Typo "Detection probability is 10% for Ops = 0.2". Ops lower bound is defined as 1.4 in eq 6, so this is likely a typo, meaning to say for Ops=2.*

This has been corrected to be in line with the corrected Ops values as discussed under main comment 4.

Detection probability is 10% for $O_{ps}$ = 0.02, 50% for $O_{ps}$ = 0.04, and 90% for $O_{ps}$ = 0.08.